# A Random Subcarrier-Selection Method Based on Index Modulation for Secure Transmission

**DOI:** 10.3390/s22072676

**Published:** 2022-03-31

**Authors:** Tao Zhan, Jiangong Chen, Shan Luan, Xia Lei

**Affiliations:** 1National Key Laboratory of Science and Technology on Communications, University of Electronic Science and Technology of China (UESTC), Chengdu 611731, China; ztzs1998@163.com (T.Z.); jg_chen1997@163.com (J.C.); 2China Academy of Space Technology (CAST), Beijing 100192, China; luanshanhrb@163.com

**Keywords:** FDA, OFDM, random subcarrier-selection, secure communication, index modulation

## Abstract

Recently, a frequency diverse array (FDA) has been employed in an orthogonal frequency division multiplexing (OFDM) transmitter to achieve secure wireless communication without mathematical encryption. However, an insecure coupling effect arises if the frequency increments are linearly assigned to all antenna elements. To solve this problem, random subcarrier-selection methods are proposed; however, the challenge lies in the random selection of subcarriers. Inspired by the randomness of index modulation (IM), this paper proposes a low complexity random subcarrier-selection method based on index modulation (RSCS-IM). Specifically, this work conducted analysis on the spectral efficiency (SE) of our system and the computational complexity of RSCS-IM, which works out a closed-form expression of the BER performance of a desired position and validates the theoretical outcomes through simulation.

## 1. Introduction

### 1.1. Problem Formulation

Wireless communication plays an important role in today’s communication systems. However, since the open nature of the environment allows illegal users to eavesdrop confidential messages, the security of wireless communication is urgently needed and therefore has attracted extensive attention and research [1,2,3].

### 1.2. Current Literature

A conventional technique for wireless-communication security is mathematical encryption [4]. Nevertheless, it suffers from heavy system overhead and computational complexity. In contrast to mathematical encryption, directional modulation (DM) is a key-less physical-layer security (PLS) transmission technique, which uses an antenna array to transmit signal only along the desired direction or even only in the desired position [5]. In the past few decades, the implementation of DM was mainly based on a phased array (PA) [6]. However, the transmit beam pattern is only angle-dependent if employing PA, and it cannot guarantee the security of the transmission when eavesdroppers share the same direction with the legitimate user.

To address such a problem, another antenna array model, a frequency diverse array (FDA), was proposed in [7]. The authors in [7] pointed out an extra range-dependent property of the FDA by introducing a frequency offset between adjacent antenna elements. Benefiting from this feature, an FDA has the range-angle dependent property and delivers potential applications in PLS. In order to generate the required beam patterns, a number of researchers have conducted investigations on the design of frequency offset [8,9,10,11,12]. Moreover, Gao et al. proposed a multi-carrier FDA scheme in [13] to improve the performance.

The carriers assigned to all antenna elements can be mutually orthogonal by designing a proper frequency offset. Thus, Ding et al. [14] constructed an orthogonal frequency division multiplexing (OFDM) system with an FDA for secure transmission. Unfortunately, as the result of the coupling effect, there were positions besides the legitimate receiver where the BER was low as well. If eavesdroppers were located at these positions, the transmission was not secure. In order to mitigate the coupling effect of a conventional FDA, the authors of [15] proposed a random FDA (RFDA) by employing random frequency increments across the array elements. Furthermore, a multi-beam wireless communication scheme was proposed in [16] with RFDA, which was security-enhanced, spectrum-efficient and power-efficient.

However, for medium-scale and large-scale DM systems, the receiver structure of RFDA dramatically increases the circuit cost, motivating the authors of [17] to propose a random subcarrier-selection array based using the OFDM technique instead of RFDA. Additionally, Shen et al. [18] gave two practical random subcarrier-selection methods: a quadratic subcarrier set plus randomization procedure (QSS Plus RP) and a prime subcarrier set plus randomization procedure (PSS Plus RP), both of which could select the subcarriers randomly and achieve secure precise wireless transmission.

### 1.3. Motivation and Related Work

It can be derived from [17,18] that the key of combining FDA with OFDM, while mitigating the coupling effect, was to randomly distribute a set of orthogonal subcarriers to each antenna array element. The authors of [18] gave two practical random subcarrier-selection methods: QSS Plus RP and PSS Plus RP. The QSS was defined as a set whose subcarrier index along the antenna array is a non-linear function of the corresponding antenna element index. The PSS was defined as a set whose subcarrier index is prime. Moreover, in order to destroy the regular order and produce a more random subcarrier index distribution over the transmitting antennas, the authors designed a block interleaving randomization procedure and repeated it until the randomization metric was larger than the predefined threshold.

However, this method brought about new problems. First, the repetition of the block interleaving operation increased system complexity, and it was hard to obtain a proper predefined threshold. Also, it generated a large number of unused subcarriers that greatly wasted spectrum.

To alleviate the above problems, this paper proposed a random subcarrier-selection method based on index modulation. On one hand, the bitstream was a random binary sequence, which could guarantee the randomness of the selected subcarriers; on the other hand, similar to IM [19], the indices of the subcarriers could also carry the information, which mitigated the waste of subcarriers. Moreover, our proposed scheme avoided the repeated block interleaving, achieving lower complexity. The proposed scheme could be used in wireless communications where the FDA is required to provide secure transmission.

### 1.4. Benefits and Challenges

The main benefits are summarized as follows:A new random subcarrier-selection method was proposed to guarantee the randomness of the selected subcarriers. In contrast to the scheme proposed in [18], RSCS-IM avoids the randomization procedure while the selected subcarriers are more random.IM is combined with subcarrier selection. By employing IM, the information was conveyed not only by MQAM modulation but also by the indices of the activated subcarriers, which improved the SE. Operating at the same SE, the BER performance was promoted by employing our scheme.The secure precise transmission via computer simulation was demonstrated and derived the closed-form expression of BER for the desired position. The theoretical outcomes were validated by simulation results as well.The main challenges are summarized as follows: in order to achieve precise and secure wireless communication, the randomness of the selected subcarriers must be guaranteed; the system complexity must be reduced and SE must be improved.

The rest of the paper can be summarized as follows. In Section 2, the system model is presented, including the conventional random subcarrier-selection method and the proposed random subcarrier-selection method based on IM. Performance analysis is depicted in Section 3. The computer simulation results are given in Section 4 and Section 5 draws the conclusion.

The notations used in this manuscript are shown in Table 1.

## 2. System Model

### 2.1. Conventional Random Subcarrier-Selection Method

The frequency increments of the conventional uniform FDA are linearly assigned to all antenna elements, resulting in an insecure coupling effect [7]. For mitigating this problem, a random subcarrier-selection method was proposed in [18], where a $N_{T}$-element antenna array was employed at the transmit side and each element transmitted the same subcarrier symbol. The used subcarriers were randomly selected from the all-subcarrier set of OFDM. Assuming that the number of total subcarriers is $N_{s}$, then we have the following:(1)$$S_{sub}=\left\{f_{m}|f_{m}=f_{c}+m\Delta f,\left(m=0,1,\ldots ,N_{s}-1\right)\right\},$$
where $f_{c}$ is the reference frequency, *m* denotes the subcarrier index, and Δf stands for the sub-channel bandwidth. The random subcarrier-selection method PSS Plus RP is described below. Firstly, the authors constructed a random subcarrier index set defined as
(2)$$S_{p}=\left\{k_{p}|k_{p}\;is\;a\;prime\;number,k_{p}\in \left(0,1,2,\ldots ,N_{s}-1\right)\right\}.$$

In order to make a more random subcarrier index distribution over the transmit antennas, the authors proposed a random procedure (RP) by prime modulo operation and block interleaving. A constant *p* was chosen, which was the largest prime less than $\sqrt{N_{T}}$. Then, the set $S_{p}$ was partitioned into *p* subsets by taking all set elements modulo *p*, which is represented as
(3)$$K=K_{0}\cup K_{1}\cup \cdots \cup K_{p-1}.$$

Finally, the authors permuted the element order in the set *K* by block interleaving [18], which is depicted in Figure 1. In addition, the values of *I* and *J* should satisfy the inequality
(4)$$\left(J-1\right)I<N_{T}<JI.$$

It is worth pointing out that the block interleaving would be repeated until the randomness of the subcarrier index was greater than a certain threshold. To measure the random degree of the selected subcarriers, a random metric (RM) $\delta_{I}$ was defined in [18]. Suppose that the indices of the selected subcarriers are represented as
(5)$$\eta =\{\eta \left(1\right),\eta \left(2\right),\cdots ,\eta \left(N_{T}\right)\}.$$

The subcarrier index spacing vector can be derived as
(6)$$\Delta \eta =[|\eta \left(1\right)-\eta \left(2\right)|,|\eta \left(2\right)-\eta \left(3\right)|,\cdots ,|\eta \left(N_{T}-1\right)-\eta \left(N_{T}\right)|].$$

Then, $\delta_{I}$ can be expressed as
(7)$$\delta_{I}=\frac{1}{N_{T}-1}\sum_{i=1}^{N_{T}-1}{\left(\Delta \eta \left(i\right)-\bar{\Delta }\eta \right)}^{2},$$
where Δη(i) is the *i*-th element of Δη. In addition, $\bar{\Delta }\eta$ denots the average value of Δη, which is given by
(8)$$\bar{\Delta }\eta =\frac{1}{N_{T}-1}\sum_{i=1}^{N_{T}-1}\Delta \eta \left(i\right),$$

Figure 2 draws the flow chart of PSS Plus RP. As shown in Figure 2, firstly, $N_{T}$ subcarriers were selected whose index was a prime number from the given subcarrier set in (Equation 1). Next, a prime number *p* was chosen, which was the largest prime less than $\sqrt{N_{T}}$. Then, these $N_{T}$ indices were partitioned into *p* groups using the modulo-*p* operation. Finally, the block interleaving operation was performed, as shown in Figure 1, on this *p*-group index and the RM was computed. If RM was larger than the predefined threshold, the index output after block interleaving was the desired index. Otherwise, the block interleaving operation was repeated until RM was larger than the predefined threshold. We could see that there were a number of unused subcarriers causing the waste of spectrum. Additionally, a proper threshold was hard to obtain and the repeated block interleaving operation increased the computational complexity.

### 2.2. The Proposed Random Subcarrier-Selection Method Based on Index Modulation

It could be seen from the description of the conventional scheme in Section 2.1 that the complexity of PSS Plus RP mainly resulted from block interleaving, which was repeated until the random degree was greater than the predefined value. Meanwhile, the spectrum was wasted since a large number of subcarriers were unused. For alleviating this problem, this paper proposed a novel random subcarrier-selection method based on index modulation (RSCS-IM) under a new FDA OFDM communication architecture. The array structure of the proposed FDA OFDM transmitter based on IM is shown in Figure 3, where the legitimate transmitter Alice equipped a $N_{T}$-element linear antenna array with element space *d*. The legitimate receiver Bob and *N* eavesdroppers Eve all employed a single antenna. In the conventional scheme [18], all antennas of Alice transmitted the same subcarrier symbol per OFDM symbol. While in our scheme, all antennas of Alice transmitted different subcarrier symbols per OFDM symbol to Bob by selecting multiple subcarriers from the subcarrier set using IM. If the first antenna was taken as the reference, Bob was located at ($\theta_{0},R_{0}$), which was represented as the red star. The positions of the *N* eavesdroppers were $\left(\theta_{1},R_{1}\right),\left(\theta_{2},R_{2}\right),\cdots ,\left(\theta_{n},R_{n}\right)$ respectively, which were represented as the black stars. Suppose there were $N_{s}$ orthogonal subcarriers in our FDA OFDM-IM system and the set of subcarriers was
(9)$$S_{sub}=\left\{f_{m}|f_{m}=f_{c}+m\Delta f,m=0,1,\ldots ,N_{s}-1\right\},$$
where $f_{c}$ is the reference frequency, *m* denotes the subcarrier index, and Δf stands for the subchannel bandwidth, $T_{s}=1/\Delta f$ denotes the period of an OFDM symbol. It was assumed that $N_{s}\Delta f\ll f_{c}$ and d=λ/2, where $\lambda =c/f_{c}$ denotes the wave length and *c* is the light speed. The subcarrier frequency assigned to the *n*-th antenna was $f_{\eta (n)}$, where $f_{\eta (n)}\in S_{sub}$ and $\eta \left(n\right)\in \left\{0,1,\cdots ,N_{s}-1\right\}$. η(n) was determined by the bits used for index selection. It is worth pointing out that for different $n_{1},n_{2}$, $\eta \left(n_{1}\right)\neq \eta \left(n_{2}\right)$. In a far-field scenario, the received signal at $\left(\theta_{0},R_{0}\right)$ could be represented by
(10)$$s\left(t\right)=\sum_{k=1}^{N_{T}}x_{k}e^{j\varphi_{k}}e^{j2\pi f_{\eta (k)}\left(t-\frac{R_{k}}{c}\right)}+\sum_{k=1}^{N_{T}}n_{k},$$
where $f_{\eta (k)}=f_{c}+\eta \left(k\right)\Delta f$, $R_{k}=R_{0}-\left(k-1\right)d\sin\theta_{0}$. $\varphi_{k}$ denotes the initial phase of the *k*-th antenna and $n_{k}$ is the received additive white Gaussian noise (AWGN) of the *k*-th antenna with the distribution of $n_{k}∼\mathrm{CN}\left(0,\sigma^{2}\right)$. So, we can rewrite (Equation 10) as
(11)$$\begin{array}{cc} s(t) & =\sum_{k=1}^{N_{T}}x_{k}e^{j\varphi_{k}}e^{j2\pi \left(f_{c}+\eta \left(k\right)\Delta f\right)\left(t-\frac{R_{k}}{c}\right)}+\sum_{k=1}^{N_{T}}n_{k}\\ \; & =\sum_{k=1}^{N_{T}}x_{k}e^{j\varphi_{k}}e^{j2\pi f_{c}t}e^{j2\pi \eta (k)\Delta ft}e^{-j2\pi f_{\eta (k)}\frac{R_{k}}{c}}+\sum_{k=1}^{N_{T}}n_{k}\\ \; & =e^{j2\pi f_{c}t}\sum_{k=1}^{N_{T}}x_{k}e^{j(\varphi_{k}-2\pi f_{\eta (k)}\frac{R_{k}}{c})}e^{j2\pi \eta (k)\Delta ft}+\sum_{k=1}^{N_{T}}n_{k}. \end{array}$$

After receiving the signal in (Equation 11), it was firstly down-converted to baseband by multiplying by $e^{-j2\pi f_{c}t}$. Then, the baseband signal was sampled at $t=n/N_{s}\Delta f$
$\left(n=0,1,\cdots ,N_{s}-1\right)$ and we obtained
(12)$$s\left(n\right)=\sum_{k=1}^{N_{T}}x_{k}e^{j(\varphi_{k}-2\pi f_{\eta (k)}\frac{R_{k}}{c})}e^{j2\pi n\eta \left(k\right)/N_{s}}+\sum_{k=1}^{N_{T}}n_{k}.$$

Observing Equation (Equation 12), if the symbols corresponding to the inactive subcarriers were set to zero and the noise was ignored, s(n) was the inverse fast Fourier transform (IFFT) of $x_{k}e^{j(\varphi_{k}-2\pi f_{\eta (k)}\frac{R_{k}}{c})}$. For the sake of demodulating the original signal correctly, the initial phase of antenna *k* should satisfy the following identity [14]
(13)$$\varphi_{k}-2\pi f_{\eta (k)}\frac{R_{k}}{c}=\varphi_{0},$$
where $\varphi_{0}$ is a constant. So, by taking the fast Fourier transform (FFT) for s(n) and multiplying by $e^{-j\varphi_{0}}$, the original signal $x_{k}$ could be restored.

The concrete random subcarrier-selection method based on index modulation (RSCS-IM) will be described in this part. It was assumed that the number of total orthogonal subcarriers and antenna array elements were $N_{s}$ and $N_{T}$, respectively. For convenience, suppose $N_{s}=2^{m_{1}}$ and $N_{T}=2^{m_{2}}$, where both $m_{1}$ and $m_{2}$ are positive integers and $m_{1}>m_{2}$. Firstly, the index of all the orthogonal subcarriers was divided into a matrix whose dimension was $N_{T}\times \frac{N_{s}}{N_{T}}$, i.e.,
(14)$$I_{sub}=\left(\begin{array}{cccc} 1 & N_{T}+1 & \cdots & N_{s}-N_{T}+1\\ 2 & N_{T}+2 & \cdots & N_{s}-N_{T}+2\\ \vdots & \vdots & \ddots & \vdots \\ N_{T} & 2N_{T} & \cdots & N_{s} \end{array}\right).$$

Next, a subcarrier was selected from every row of $I_{sub}$, respectively. For every row, *p* bits were used to select the subcarrier, where
(15)$$p=log_{2}\frac{N_{s}}{N_{T}}=m_{1}-m_{2}.$$

Figure 4 depicts the flow chart of the proposed scheme, RSCS-IM. As shown in Figure 4, firstly, all the indices from the orthogonal subcarrier set in (Equation 9) were divided into the matrix in (Equation 14). Next, $p\times N_{T}$ bits were generated and each *p*-bit of binary data was converted to a decimal, according to the data bits generated by the source, where $p=log_{2}\frac{N_{s}}{N_{T}}$. Finally, one index from every row of matrix (Equation 14) was picked in turn using these $N_{T}$ decimal data as columns and we obtained the $N_{T}$ desired indices. So, we could see that the proposed scheme avoided the repeated block interleaving operation and guaranteed the randomness of the selected subcarriers. Moreover, the indices of the activated subcarriers could transmit extra data bits, mitigating the waste of spectrum.

Hence, the whole block diagram of FDA OFDM-IM system is described in Figure 5, where $m=\left(p+q\right)N_{T},p=m_{1}-m_{2},q=log_{2}M$ and *M* is the order of modulation.

As shown in Figure 5, a total of *m* information bits entered the bit splitter. These *m* bits were split into $2N_{T}$ groups. For the first $N_{T}$ groups, each group contained *p* bits for index-selection while the other $N_{T}$ groups each contained *q* bits for mapping. After that, the signal was conveyed by the FDA OFDM-IM system in the far-field scenario and Bob received the signal s(t) in (Equation 10). Just as depicted above, taking down-conversion and sampling for s(t) in turn, we could obtain s(n) in (Equation 12). Next, an $N_{s}$-point FFT was obtained for s(n). Then, in order to detect the activated subcarriers, we adopted the scheme of greedy detection and determined the location of maximum energy for every row in (Equation 14) as the index of the activated subcarrier. Finally, the index demodulation and QAM demodulation were obtained, respectively, and the original data could be restored. As for the eavesdroppers, assuming that they were consistent with the demodulation method used by the legitimate receiver, from (Equation 13) we could see that there was a phase offset between the received data and original data. So they could not restore the original data correctly.

In summary, the proposed scheme could guarantee the randomness of the selected subcarriers because of the randomness of the information bits. Moreover, the indices of the activated subcarriers could transmit extra data bits, improving the SE. In Section 3, we will analyse the performance of RSCS-IM and PSS Plus RP in detail.

## 3. Performance Analysis

In this section, we analyse some important factors of the proposed scheme RSCS-IM, including the SE, complexity and BER performance of the desired position. Meanwhile, the factors mentioned above of the scheme’s “prime subcarrier set plus randomization procedure (PSS Plus RP)” proposed in [18] were also analyzed for comparison.

### 3.1. Spectral Efficiency and Computational Complexity Analysis

Spectral Efficiency AnalysisOne OFDM symbol was taken for reference. According to the principle of RSCS-IM, an OFDM symbol included $N_{1}$ bits information, which was defined as
(16)$$N_{1}=N_{T}log_{2}\frac{N_{s}}{N_{T}}+N_{T}log_{2}M,$$
where $N_{T}$ is the number of antennas, $N_{s}$ represents the number of total subcarriers and *M* is the modulation order. While for PSS Plus RP, the counterpart was
(17)$$N_{2}=N_{T}log_{2}M.$$It was obvious that $N_{1}-N_{2}=N_{T}log_{2}\frac{N_{s}}{N_{T}}>0$, so RSCS-IM promoted the spectrum efficiency. Meanwhile, we could see that a large $N_{s}$ implied a higher spectrum efficiency; however, larger $N_{s}$ also implied higher computational complexity. We could select the value of $N_{s}$ and $N_{T}$ to meet our requirement in a certain scenario.Computational Complexity AnalysisAccording to the principles of PSS Plus RP and RSCS-IM shown in Figure 2 and Figure 4, respectively, we analyze the computational complexity of these two schemes in this subsection. The computational complexity was evaluated in terms of the real-valued operations, including real-valued multiplication, real-valued additions, and real-valued modulo operations. According to the principles, we gave the complexity of these two methods shown in Table 2, where $N_{s}$ and $N_{T}$ denote the number of total subcarriers and antennas, respectively, and *N* is the number of loops of block interleaving.

From Table 2 we could clearly see that the number of modulo operations in RSCS-IM was less than that of PSS Plus RP. As for the addition operation, in general, $N_{T}<4N$ and $log_{2}\frac{N_{s}}{N_{T}}<N_{T}$, so $N_{T}\left(log_{2}\frac{N_{s}}{N_{T}}-1\right)<4N\left(N_{T}-1\right)$. Moreover, $log_{2}\frac{N_{s}}{N_{T}}<10$ in most cases. $log_{2}\frac{N_{s}}{N_{T}}\left(log_{2}\frac{N_{s}}{N_{T}}-1\right)/2$ and *N* had the same order of magnitude. Thus, we could see that the number of multiplications in these two schemes were basically equal. Overall, the complexity of RSCS-IM was lower.

### 3.2. BER Performance Analysis of the Desired Position

In what follows, the average BER of the desired position is analyzed. As shown in Figure 5, demodulation includes index demodulation and QAM demodulation, both of which need to know the index of the activated subcarriers. For all alternative orthogonal subcarriers, we could consider that the activated subcarriers conveyed the QAM symbol plus noise while the inactivated subcarriers just conveyed the noise. As a result, we adopted greedy detection to determine the index of activated subcarriers. Suppose that $y_{i}\left(\alpha \right)$ denotes the symbol whose index is α of the *i*-th row in (Equation 14), then,
(18)$$\hat{\alpha }=\underset{\alpha }{\arg\max}{\left|y_{i}\left(\alpha \right)\right|}^{2}.$$

After determining the index of the activated subcarriers, ML detection was adopted to determine the constellation symbols, which was similar to the classical QAM demodulation, i.e.,
(19)$$\hat{s}\left(\hat{\alpha }\right)=\underset{x(\hat{\alpha })\in S}{\arg\min}{\left|y\left(\hat{\alpha }\right)-x\left(\hat{\alpha }\right)\right|}^{2}.$$
where *S* is the set of standard constellation symbols and $y(\hat{\alpha })$ denotes the received symbol.

According to the analysis above, average BER could be represented as
(20)$$P_{s}=\frac{n_{i}+n_{Q}}{n_{t}},$$
where $n_{i}$ and $n_{Q}$ denote the number of wrong bits of index demodulation and QAM demodulation, respectively. $n_{t}$ denotes the total bits of an OFDM symbol, including the bits of index selection. As the index selection of each group, i.e., each row in (Equation 14) did not affect each other, taking an arbitrary group for example, we could get
(21)$$n_{t}=log_{2}\left(\frac{N_{s}}{N_{T}}\right)+log_{2}M.$$

It was assumed that $P_{g}$ denoted the error probability of greedy detection. When greedy detection was wrong, the bit error rate of index was $P_{i}$ and the counterpart of QAM modulation was $P_{Q1}$. Otherwise, the bit error rate of QAM modulation was $P_{Q2}$. So, $n_{i}$ could be written as
(22)$$n_{i}=P_{g}P_{i}log_{2}\left(\frac{N_{s}}{N_{T}}\right),$$
and $n_{Q}$ as
(23)$$n_{Q}=P_{g}P_{Q1}log_{2}M+\left(1-P_{g}\right)P_{Q2}log_{2}M.$$

From the principal of greedy detection, when greedy detection was wrong, i.e., $max\{|n\left(\hat{\alpha }\right){|}^{2}\}>|x\left(\alpha \right)+n\left(\alpha \right){|}^{2}$, we have
(24)$$P_{g}=P(max\{|n\left(\hat{\alpha }\right){|}^{2}\}>|x\left(\alpha \right)+n\left(\alpha \right){|}^{2}).$$

In line with [20], a closed-form expression of (Equation 24) is
(25)$$P_{g}=1-\sum_{k=0}^{\frac{N_{s}}{N_{T}}-1}\left(\begin{array}{c} N_{s}/N_{T}-1\\ k \end{array}\right)\frac{{\left(-1\right)}^{k}}{k+1}e^{\gamma_{\alpha }\left(\frac{1}{k+1}-1\right)},$$
where $\gamma_{\alpha }$ denotes the SNR of activated subcarriers and $\left(\begin{array}{c} N_{s}/N_{T}-1\\ k \end{array}\right)$ is the binomial coefficient. When it came to $P_{i}$, we knew that the index demodulation was turning the decimal index into a binary bit stream and the index met $I\in [0,\frac{N_{s}}{N_{T}}-1]$. The essence of wrong greedy detection was detecting the correct index as one of the other $\frac{N_{s}}{N_{T}}-1$ wrong indexes. Meanwhile, the probability of detecting the correct index as any other wrong index was the same. So,
(26)$$P_{i}=\frac{\frac{1}{2}log_{2}\left(\frac{N_{s}}{N_{T}}\right)\frac{N_{s}}{N_{T}}}{\left(\frac{N_{s}}{N_{T}}-1\right)log_{2}\left(\frac{N_{s}}{N_{T}}\right)}=\frac{\frac{N_{s}}{N_{T}}}{2(\frac{N_{s}}{N_{T}}-1)}.$$

As for $P_{Q1}$ and $P_{Q2}$, after a brief analysis, we could get
(27)$$P_{Q1}=\frac{1}{2},$$
and
(28)$$P_{Q2}=P_{QAM},$$
where $P_{QAM}$ denotes the BER of classical QAM modulation. Hence, substituting (Equation 25)–(Equation 28) into (Equation 22) and (Equation 23) we could obtain $n_{i}$ and $n_{Q}$. Finally, $P_{s}$ could be derived by substituting (Equation 21)–(Equation 23) into (Equation 20).

In terms of PSS Plus RP, it did not include the index demodulation and its BER performance was similar to the classical OFDM system in the Gaussian channel, which could be represented as
(29)$$P=P_{QAM}.$$

From (Equation 28) and (Equation 29) we could see that $P_{s}$ was equal to *P* when the greedy detection was always right, i.e., $P_{i}=0$, which was in line with the analysis.

In terms of Eve, it is worth pointing out that, when employing an RSCS-IM scheme, the bits transmitted by te index could be demodulated correctly when SNR was high enough i.e.,
(30)$$P_{e1}=\frac{\frac{1}{2}log_{2}M}{log_{2}\left(\frac{N_{s}}{N_{T}}\right)+log_{2}M},$$
where $P_{e1}$ represents the BER of Eve when SNR was high enough for RSCS-IM. While for PSS Plus RP, assuming $P_{e2}$ represents the BER of Eve, regardless of the value of SNR,
(31)$$P_{e2}=\frac{1}{2}$$
since it did not include the index modulation.

## 4. Simulation Results

In this section, the simulation results of the random degree, the computational complexity, and the BER performance are depicted to describe the performance of RSCS-IM. In this simulation, system parameters were chosen as follows: reference frequency $f_{c}=2.404$ GHz [14], signal bandwidth B=20 MHz, the number of total subcarriers $N_{s}=512$ or 1024, $E_{b}/N_{0}=9$ dB, the number of antenna array elements $N_{T}=64$, the legitimate receiver Bob was located at ($30^{\circ },100$ m), and we employed a QPSK constellation. The constant $\varphi_{0}=0$ in (Equation 13). These parameters are listed in Table 3.

### 4.1. The Simulation Results of Random Degree

To measure the random degree of the selected subcarriers of the proposed RSCS-IM and the PSS Plus RP [18], we calculated $\delta_{I}$ in (Equation 7) of the two random subcarrier-selection methods many times. The results are presented in Figure 6. As shown in Figure 6, the random degree of the proposed scheme RSCS-IM was always larger than that of PSS Plus RP in ten thousand simulations, meaning that the selected subcarriers employing RSCS-IM were more random. Additionally, comparing Figure 6a with Figure 6b, we could see that when $N_{s}=1024$, the random degree was improved using both methods because the number of available subcarriers increased. Moreover, the increase in random degree was greater when employing RSCS-IM rather than employing PSS Plus RP since the number of increased subcarriers employing RSCS-IM was more than the number of increased primes employing PSS Plus RP.

### 4.2. The Simulation Results of Computational Complexity

To visualize the computational complexity of the two schemes, we simulated the number of addition, multiplication, and modulo operations and selected $N_{s}=512/1024$ and $N_{T}=64$. When $\delta_{I}$ firstly reached 95% of the maximum random degree, the number of loops N=23/22. The concrete computational complexity is depicted in Figure 7. Hence, we could see that the number of additions, multiplications, and modulo operations in RSCS-IM were all less than that of PSS Plus RP. The complexity of RSCS-IM was lower. Comparing Figure 7a with Figure 7b, when $N_{s}=1024$, the complexity of PSS Plus RP was basically unchanged since the number of loops was basically independent of $N_{s}$, while the complexity increased slightly, when employing RSCS-IM, in addition and multiplication operations. However, the complexity of RSCS-IM was still substantially lower than that of PSS Plus RP when $N_{s}$ was not particularly large.

### 4.3. The Simulation Results of BER Performance

In order to evaluate the secure performance of the proposed scheme RSCS-IM, the 3-D performance surface of BER versus the direction angle and distance was simulated. Furthermore, we simulated the performance of PSS Plus RP to make a comparison with our scheme. Meanwhile, in order to illustrate the coupling effect of a uniform FDA, and to further illustrate the importance of randomly selecting subcarriers, this paper also simulated the 3-D performance surface of BER versus the direction angle and distance of uniform FDA [7].

Figure 8 is the 3-D performance surface of BER versus the direction angle and distance of a uniform FDA. As shown in Figure 8, besides the desired position, there were other positions where the BER was low as well. If eavesdroppers were located at these positions, the transmission was not secure. This phenomenon is known as the coupling effect of a uniform FDA.

Figure 9 depicts the 3-D BER performance surface versus the direction angle and distance when the number of total subcarriers $N_{s}=512$. Compared with Figure 8, Figure 9 depicts that low bit error rate occurred only in a small area around the desired position. The other positions could not restore the original signal correctly. Therefore, the random selection of subcarriers was important for weakening the coupling effect and, thus, achieving safe and accurate wireless transmission.

When $N_{s}=1024$, Figure 10 depicts the 3-D BER performance surface versus the direction angle and distance. Figure 10a is similar to Figure 9a. Comparing Figure 10b with Figure 9b, the BER performance at Eve was slightly improved because of the increased weight of bits used for index selection. In terms of the desired position, there was a certain BER gain when employing RSCS-IM, comparing Figure 10b with Figure 10a.

Figure 11 depicts the BER performance versus $E_{b}/N_{0}$ of Bob. As shown in Figure 11, the theoretical and simulation curves were basically consistent. Specifically, regardless of $N_{s}=512$ or $N_{s}=1024$, the BER performance of Bob was consistent when employing PSS Plus RP since it did not include IM. The BER performance of Bob was only related to the symbol modulation method. Additionally, when $N_{s}=512$ and BER was $10^{-2}$, there was more than 1dB gain when employing RSCS-IM rather than PSS Plus RP. When $N_{s}=1024$, there was an extra gain in the BER performance of Bob compared with $N_{s}=512$ in high $E_{b}/N_{0}$, since SE was improved. While in low $E_{b}/N_{0}$, there was a distinct degradation in the error probability of greedy detection. So the BER performance of Bob was basically consistent in low $E_{b}/N_{0}$ when $N_{s}=512$ and $N_{s}=1024$. Likewise, since the error probability of greedy detection was very high in low $E_{b}/N_{0}$, the BER performance of Bob employing RSCS-IM was worse than employing PSS Plus RP.

From Figure 9 we can see that the BER performance of the other positions, other than the desired position, was of small variation. So, when taking an arbitrary position ($60^{\circ },50$ m) as an eavesdropper (Eve), the BER performance versus SNR using RSCS-IM with different $N_{T}$ is shown in Figure 12. The number of total subcarriers $N_{s}=512$. As for Eve, the BER performance versus SNR was always about $\frac{1}{2}$ when employing PSS Plus RP. As the other positions could demodulate the index bits correctly, the BER performance employing RSCS-IM was less than $\frac{1}{2}$. Specifically, the bigger $\frac{N_{s}}{N_{T}}$ was, the lower the BER was in high SNRs. In other words, the less bits used in IM, the higher the BER was in high SNRs. Thus, we could control the BER performance in other positions by selecting a proper value of $\frac{N_{s}}{N_{T}}$ so that Eve could not restore the original signal correctly, even if error correction coding was employed.

## 5. Conclusions

Based on IM, this paper proposed a new random subcarrier-selection method, RSCS-IM, which achieved secure and precise wireless transmission. Compared to PSS Plus RP, which was proposed in [18], RSCS-IM not only enhanced the SE but also reduced complexity. At the same SE, the proposed RSCS-IM scheme featured better BER performance. Moreover, the BER performance of an eavesdropper could be limited by selecting a proper value of $N_{s}/N_{T}$. Finally, we derived the closed-form expression of BER for the desired position and demonstrated the performance of our scheme via numerical simulation.

## Figures and Tables

**Figure 1 sensors-22-02676-f001:**
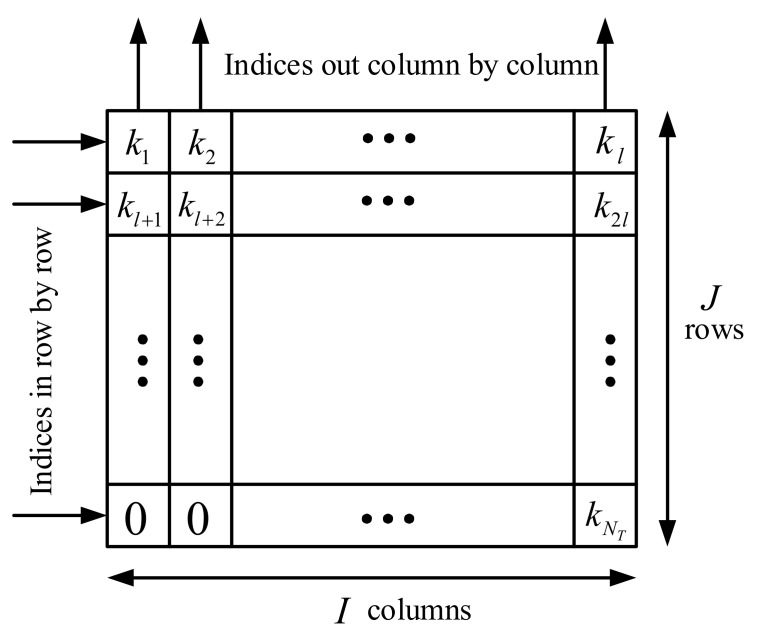
The principle of block interleaving.

**Figure 2 sensors-22-02676-f002:**
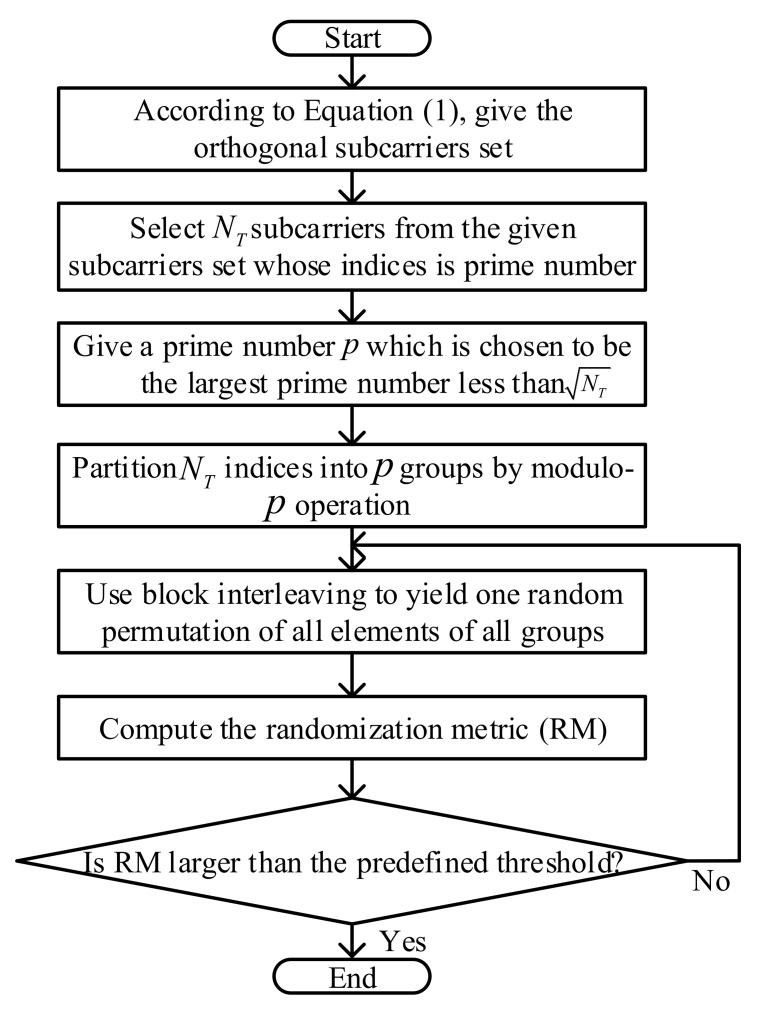
Flow chart of PSS Plus RP.

**Figure 3 sensors-22-02676-f003:**
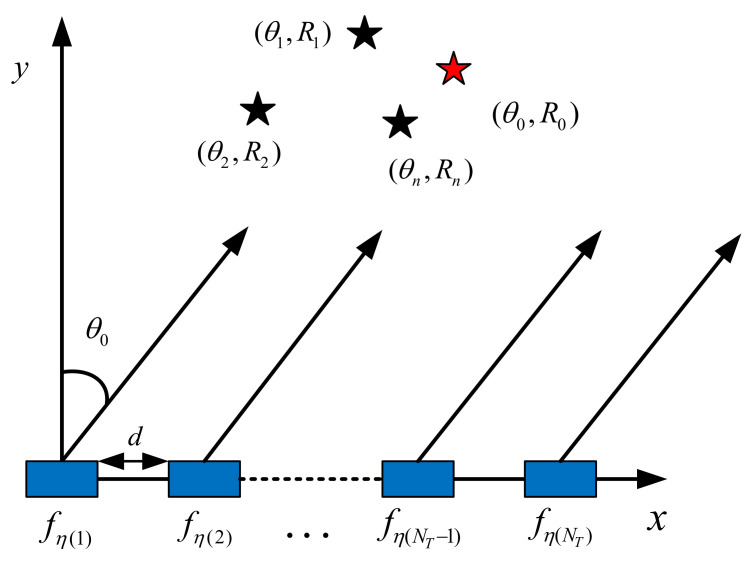
Array Structure of the FDA OFDM-IM Transmitter.

**Figure 4 sensors-22-02676-f004:**
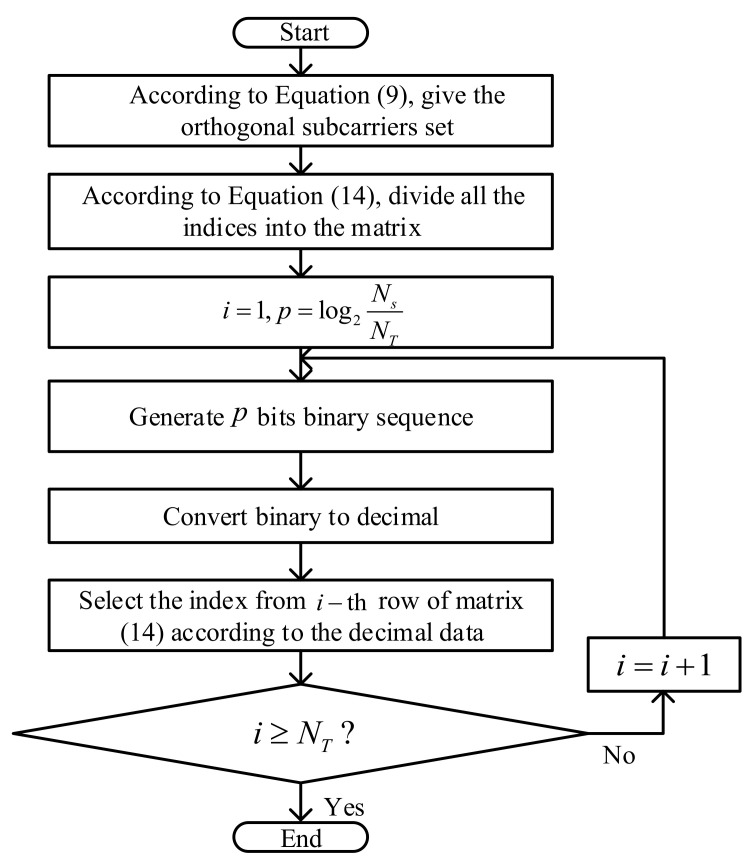
Flow chart of RSCS-IM.

**Figure 5 sensors-22-02676-f005:**
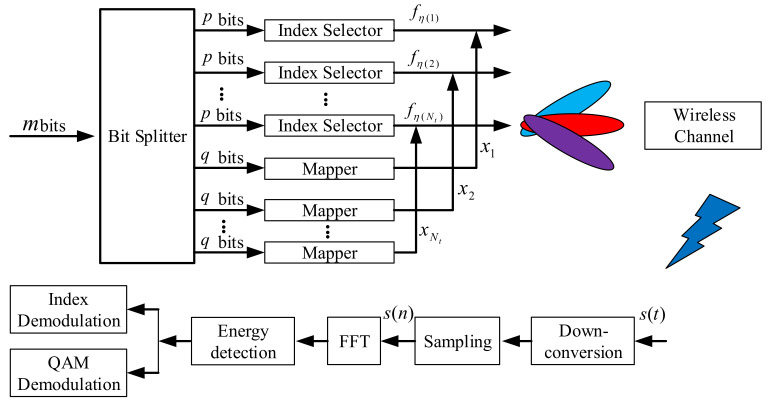
Block diagram of the FDA OFDM-IM system.

**Figure 6 sensors-22-02676-f006:**
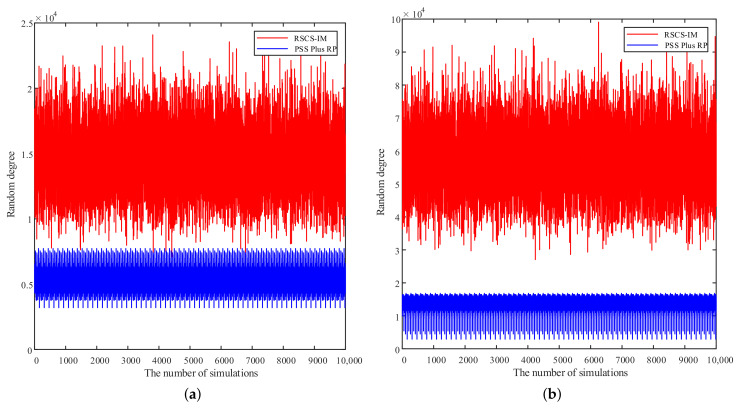
Simulation comparison of the random degree. (**a**) $N_{s}=512.$ (**b**) $N_{s}=1024$.

**Figure 7 sensors-22-02676-f007:**
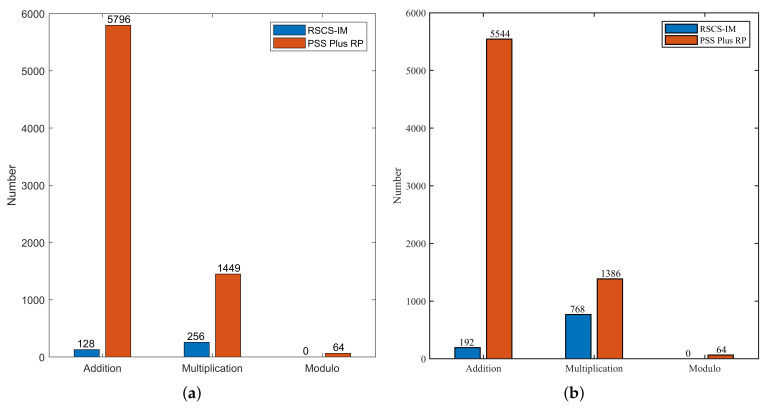
The computational complexity of RSCS-IM and PSS Plus RP. (**a**) $N_{s}=512.$ (**b**) $N_{s}=1024$.

**Figure 8 sensors-22-02676-f008:**
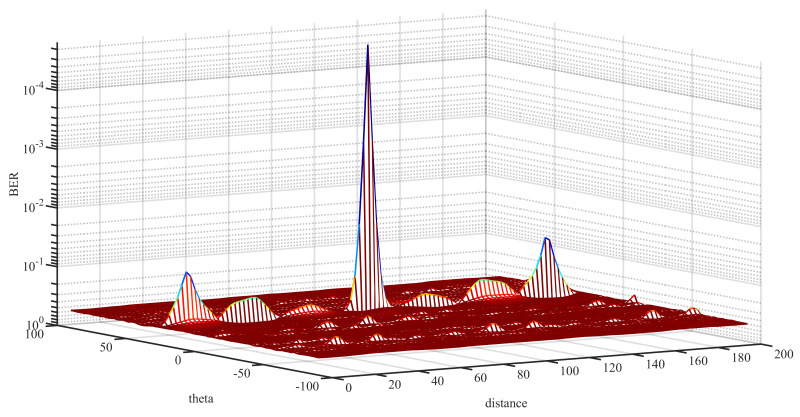
3-D performance surface of a uniform FDA.

**Figure 9 sensors-22-02676-f009:**
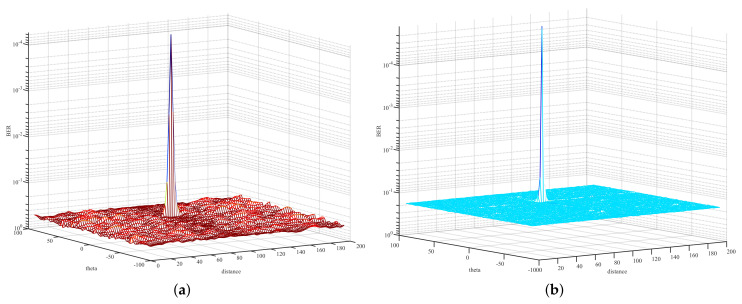
3-D performance surface of BER versus the direction angle and distance ($N_{s}=512$). (**a**) 3-D performance surface of BER using PSS Plus RP; (**b**) 3-D performance surface of BER using RSCS-IM.

**Figure 10 sensors-22-02676-f010:**
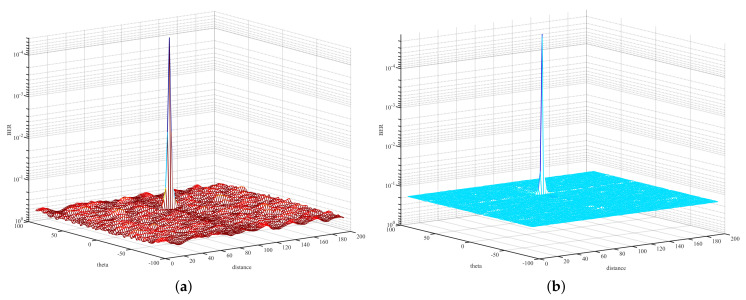
3-D performance surface of BER versus the direction angle and distance ($N_{s}=1024$). (**a**) 3-D performance surface of BER using PSS Plus RP; (**b**) 3-D performance surface of BER using RSCS-IM.

**Figure 11 sensors-22-02676-f011:**
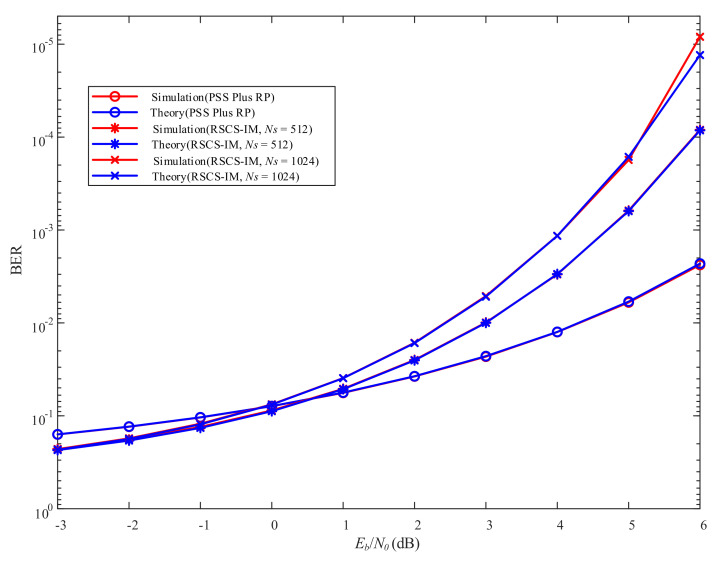
BER performance versus $E_{b}/N_{0}$ of Bob.

**Figure 12 sensors-22-02676-f012:**
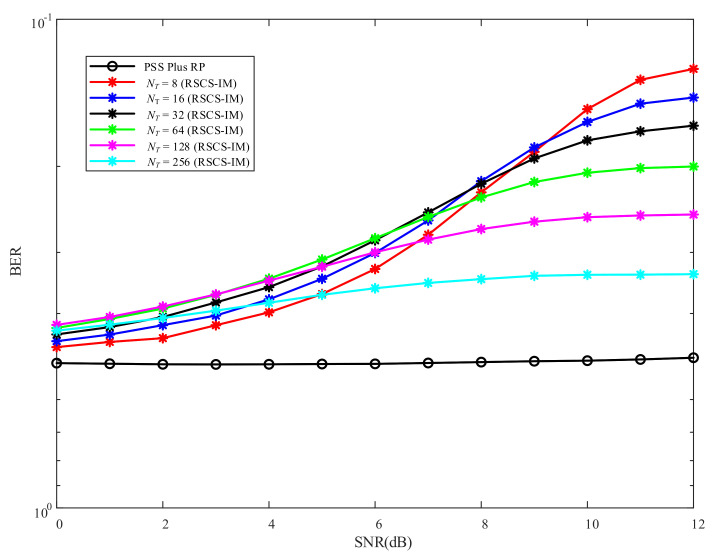
BER performance versus SNR of Eve under RSCS-IM and PSS Plus RP schemes with different $N_{T}$.

**Table 1 sensors-22-02676-t001:** Notations used in this manuscript.

Notation	Description
∑	Sum.
$X∼\mathrm{CN}(0,\sigma^{2})$	The distribution of a circularly symmetric complex Gaussian.
$\left(\begin{array}{c} n\\ k \end{array}\right)$	The binomial coefficient.

**Table 2 sensors-22-02676-t002:** Complexity of PSS Plus RP and RSCS-IM.

Scheme	Complexity		
	Addition	Multiplication	Modulo
RSCS-IM	$N_{T}\left(log_{2}\frac{N_{s}}{N_{T}}-1\right)$	$\frac{N_{T}log_{2}\frac{N_{s}}{N_{T}}\left(log_{2}\frac{N_{s}}{N_{T}}-1\right)}{2}+N_{T}$	—
PSS Plus RP	$4N(N_{T}-1)$	$N(N_{T}-1)$	$N_{T}$

**Table 3 sensors-22-02676-t003:** The system parameters in our simulation.

Parameter	Value	Description
$f_{c}$	2.404 GHz	Reference frequency.
*B*	20 MHz	Signal bandwidth.
$N_{s}$	512/1024	The number of total subcarriers.
$N_{T}$	64	The number of antenna array elements.
$E_{b}/N_{0}$	9 dB	Average bit energy to noise power ratio.
($\theta_{0},R_{0}$)	($30^{\circ },100$ m)	The desired position.
$\varphi_{0}$	0	The constant of Equation (Equation 13).

## Data Availability

Not applicable.

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
