# Peer review of "A Random Subcarrier-Selection Method Based on Index Modulation for Secure Transmission"

_sensors, 2022, doi:10.3390/s22072676_

Round 1

Reviewer 1 Report

There is many major concern with this paper and below few comments:

-Few abbreviations are undefined, e.g., RSCS-IM
-Punctuation error can be noticed, e.g., bit error rate(BER)
-Few terms have been used many times, but they are not abbreviated.

e.g., spectral efficiency should be abbreviated as spectral efficiency (SE) 
and rest repetitions to be replaced by SE.

-Introduction needs to be improved to a different level by incorporating the following points:
i) do literature survey based on recently published papers since 2021, 2020, 2019 and few relevant citations should be from MDPI journals.
ii) add a different sub-section such as `motivation and related work'
iii) applications of the proposed scheme
iv) benefits
v) challenges

-`Figure 4. Block diagram for PSS Plus RP.' : Block diagram-->flow chart

Likewise caption of fig 5 to be changed.

-Flow charts are not well described at all.

-Scaling of BER should start with 10^0, which means this should follow as 10^0 to 10^{-4} instead of  10^{-4} to 10^0.

All the BER figures are associated with two schemes such as `RSCS-IM' and `PSS Plus RP[8]', but figure 9 does not reflect any scheme.

-All the figures need  detail discussions and explanations. Provide separate paragraph for each figure.

-In your paper, the main contribution is the proposed `RSCS-IM scheme' but unfortunately this part has not well investigated.
Infact the proposed scheme (section 3) does not connect well with the system model (section 2).

-Figures 7 and 8 should combine!!

-Figures are not impressive. Only few results presented. Present few results considering other parameters.

-Very poor analytical part specially with the proposed scheme `RSCS-IM'. 

-Unable to find any exciting part.

-Very bad organization and presentation. Equations need an extra care.

-Add two tables, one for symbols/notation and another for simulation settings and all.

Reviewer 2 Report

The presented work is interesting and novel. The main drawback of the paper is relared to the presentation format. The introductory part is mostly refering to the state of the art in the area and not the main challenges of the proposed work. I would suggest to the authors to re-edit the introduction keeping the basic challenges and motivation for their work. Following, a state of the art and beyond section can be introduced, detailed the existing work on the area and the studies where the current work is based. A subsection can be associated with their contribution on the study. This will help a lot to follow the rest of the paper, where the overall design and implementation is detailed.

Round 2

Reviewer 1 Report

The revised version improved a lot through review comments, but some of the comments are partially addressed or not well addressed 
or may be not addressed at all:

1. Introduction needs to be improved to a different level by incorporating the following points:
i) do literature survey based on recently published papers since 2021, 2020, 2019 and few relevant citations should be from MDPI journals.
ii) add a different sub-section such as `motivation and related work'
iii) applications of the proposed scheme
iv) benefits
v) challenges

This comment has not well addressed!!

2. Flow charts are not well described at all.

This comment has not been addressed at all.

3. Scaling of BER should start with 10^0, which means this should follow as 10^0 to 10^{-4} instead of  10^{-4} to 10^0.

Partially addressed, need to work for Figs 10 and 11.

4. Very poor analytical part specially with the proposed scheme `RSCS-IM'. 

Attempt has not done at best, need to work on this comment.

5. Add two tables, one for symbols/notation and another for simulation settings and all.

A table for symbols/notations has not been added.

Reviewer 2 Report

My main concern with the paper is still its readability. My main comment in the first review has not been tackled. I'm adding it again for consideration:

"I would suggest to the authors to re-edit the introduction keeping the basic challenges and motivation for their work. Following, a state of the art and beyond section can be introduced, detailed the existing work on the area and the studies where the current work is based. A subsection can be associated with their contribution on the study."

Furthermore, the list of abbreviations have to be moved to the end of the paper, according to the template.
